# The Relationship between Perceived Organizational Support, Work Engagement, Organizational Citizenship Behavior, and Customer Orientation in the Public Sports Organizations Context

**DOI:** 10.3390/bs14030153

**Published:** 2024-02-21

**Authors:** Jongchul Park, Jooyoung Kim

**Affiliations:** 1Department of Community Sport, Korea National Sport University, Seoul 05541, Republic of Korea; righteous26@naver.com; 2Department of Sport Industry, Korea National Sport University, Seoul 05541, Republic of Korea

**Keywords:** public sports organization, perceived organizational support, work engagement, organizational citizenship behavior, customer orientation

## Abstract

As sports activities have recently become socio-culturally important in South Korea, the roles and functions of public sports organizations are attracting attention. In this situation, perceived organizational support is considered one of the significant variables to explain the attitudes and behaviors of employees within the organizations. Hence, the purpose of this study is to investigate the relationship between perceived organizational support of public sports organizations, work engagement, organizational citizenship behavior, and customer orientation and examine the mediating effect of work engagement. This study collected data from 248 employees working for public sports organizations, and data were analyzed with SPSS 26.0 and AMOS 26.0. The results showed the following. (1) Perceived organizational support has a significant positive effect on work engagement but does not affect organizational citizenship behavior and customer orientation. (2) Work engagement significantly positively affects organizational citizenship behavior and customer orientation. (3) Work engagement has been shown to fully mediate the relationship between perceived organizational support, organizational citizenship behavior, and customer orientation. This study suggests that public sports organizations need an efficient support strategy that can maximize employees’ work engagement. For example, organizations should increase their sense of unity with employees and understand the importance of their work to strengthen perceived organizational support. Lastly, organizations need to create an environment where employees can devote themselves to and focus on their work.

## 1. Introduction

As the public’s desire and expectations for participation in sports activities in South Korea increase, the value and role of public sports organizations (PSOs) are becoming increasingly important. PSOs carry out various projects and tasks to promote and revitalize elite sports and sports for all [1]. In addition, PSOs, such as the Korea Sports and Olympic Committee (KSOC) and Korea Sports Promotion Foundation (KSPO), ultimately seek to improve people’s health, happiness, and social integration [2,3]. At this time, since employees of public organizations have a profound effect on public satisfaction with administrative services [4], PSOs should strengthen their efforts to motivate them and improve work performance.

Yet, the bureaucratic structure of public organizations has been pointed out as a key factor that hinders organizational performance [5]. Rainey [6] argued that public organizations have more complex guidelines, red tape, and hierarchical structures than private organizations. Similarly, PSOs in South Korea are known to have bureaucratic properties with a formal work style and a strong control tendency [1]. Also, public organizations have a structure in which it is difficult to quantify organizational performance as individual performance, making it difficult to demonstrate individual capabilities [7]. In other words, PSOs should continue encouraging their employees to carry out active and customer-oriented tasks. Considering that the quality of sports administrative services may vary depending on the organization’s level of support, improving perceived organizational support (POS) should be preceded.

POS indicates an overall awareness of the degree to which an organization values the contribution and welfare of its employees [8]. A high level of POS generally increases positive work attitudes and belonging to the organization, eventually improving work engagement [9]. Furthermore, organizational support has a positive effect on the extra-role behavior of employees and the formation of reciprocal attitudes toward the organization [10,11]. Similarly, POS has a positive effect on the customer orientation of employees [12]. Customer orientation (CO) means that employees of an organization strive to provide administrative services that meet customer needs [13]. CO varies depending on how the employees are treated by the manager, which can lead to customer satisfaction [14,15]. Thus, CO is significant as it can ultimately result in the satisfaction of various stakeholders in the quality of sports administration services. Meanwhile, POS directly affects employees’ work engagement (WE) [16]. Employees with high levels of WE tend to have a strong desire to achieve challenging goals and devote a lot of power and effort to their work [17,18]. This means that the organization’s WE can play an important bridge role in bringing out job performance [19], so PSOs need to improve the WE of their employees.

Regarding prior research on POS in sports in South Korea, the focus is mainly on studies on private sports centers [20,21,22,23,24]. The studies generally mentioned that POS positively affected job satisfaction, organizational commitment, CO, and organizational citizenship behavior (OCB). Furthermore, Keshtgar, Naghshbandi, and Nobakht [25] reported that POS by the Department of Sports in the government directly or indirectly affects organizational mission statements and organizational performance. In particular, Prymakova and Lallatin [26] recently stated that POS has a positive effect on employees’ commitment, motivation, satisfaction, and stress reduction due to a meta-analysis of POS of public or non-profit organizations. As such, various studies recognize the importance of POS, but the studies on POS by employees of PSOs are insignificant. If employees who provide public sports services do not feel a sense of organizational support, it would not be easy to satisfy various stakeholders or actively act within the organization. This study is a fundamental approach to strengthening organizational effectiveness through the POS of employees and creating a sustainable job performance environment. This study is meaningful as organizations can identify a virtuous cycle structure that can evoke voluntary and customer-oriented behavior accompanied by employees’ passion and dedication.

In this sense, this study aims to identify the relationship between POS, WE, OCB, and CO in PSOs and to investigate the mediating effect of WE. This study is expected to contribute to efficient organizational management and improve the quality of public sports administration services by encouraging voluntary and customer-oriented behavior based on the commitment and enthusiasm of employees for their jobs. Lastly, this study aims to present useful theoretical and practical implications for establishing effective human resource management strategies for PSOs in South Korea.

## 2. Literature Review and Research Hypothesis

### 2.1. Perceived Organizational Support (POS) and Work Engagement (WE)

POS represents an employee’s overall belief in the degree to which an organization values the employees’ contribution and shows interest in their welfare [8]. POS is related to listening to employees’ complaints and trying to help when faced with difficulties [8]. In other words, POS is explained by their comprehensive perception of whether the organization fairly treats its employees. POS is based on social exchange theory and reciprocity norms [27]. In particular, the discussion of the social exchange theory of POS was carried out in earnest by Eisenberger et al. [28]. The core of this theory is that the organization’s favor to its employees will cause their compensation psychology to the organization. Thus, an employee’s high level of POS can result in a high level of attachment, commitment, and OCB to the organization [29]. POS is formed by recognition of employee performance, interest in job satisfaction, and organizational support for competency development [30]. Several previous studies have divided the components of POS into socio-emotional support and instrumental support [31]. Socio-emotional support refers to intangible support in which an organization respects and recognizes the dedication and contribution of employees, while instrumental support refers to material support necessary for employees’ job performance [32]. Considering that public organizations have limited material support in reality compared to private organizations, this study focused on socio-emotional support among POS.

POS could positively strengthen organizational commitment and the job performance capability of employees [33]. When employees receive support from the organization, POS increases and can positively affect organizational commitment and performance [34,35]. Furthermore, POS helps increase mental awareness, a positive job attitude, and a sense of belonging to the organization and job [9]. In particular, POS can strengthen the employees’ intrinsic interest in their jobs [36]. This is because POS can improve employees’ self-efficacy and act as a driver of job commitment [37,38]. In fact, it was found that employees are seen to be more engaged with their work in organizations with a high level of support [36]. As such, POS seems closely related to the WE of employees, which means commitment, passion, and dedication to the job. The causal relationship between the two concepts can be inferred in that POS is a motivating factor, and WE refers to a motivated psychological state [8,39]. As a result, POS creates an environment in which employees can feel WE with positivity [40]. Previous studies also showed that employees’ POS positively affected WE and organizational commitment, supporting the causal relationship between the two variables [16,41,42,43,44,45]. In this context, PSO employees are likely to contribute positively to WE. Based on existing theoretical and previous empirical studies, this study proposes the following hypothesis:

**Hypothesis 1 (H1).** 
*POS in PSOs has a positive effect on the WE of employees.*


### 2.2. Perceived Organizational Support (POS), Organizational Citizenship Behavior (OCB), and Customer Orientation (CO)

POS can have a positive effect on the OCB and the CO of employees. Basically, when an organization cares about and respects its employees, they have affection for the organization and try to contribute to the achievement of the organization’s purpose [43]. In this regard, it was argued that employees who highly perceive organizational support perform a lot of OCB [28], and POS by employees is considered one of the most important outcome variables of OCB [44]. Organ [45] noted that OCB comprises five factors: altruism, courtesy, conscientiousness, civic virtue, and sportsmanship. The five components of OCB can be defined as follows [46]. Altruism means helping colleagues struggling with work within the organization, and courtesy is related to activities that prevent work-related conflicts with others. Conscientiousness refers to the discretionary behavior of employees above the level of the role expected by the organization, and civic virtue indicates that employees act with a deep interest in the organization. Sportsmanship is related to the attitude of persevering in dissatisfaction and complaints toward the organization and conforming to the organization’s environment. In this study, altruism, courtesy, conscientiousness, civic virtue, and sportsmanship proposed by Organ [45] are composed of OCB expected of employees of PSOs.

Meanwhile, previous studies reported that POS had a positive effect on extra-role behavior or OCB [10,47], demonstrating a close relationship between the two variables. This means that if employees think the organization treats them, they are willing to expand their areas of activity, help their colleagues, and act voluntarily to benefit the organization. It means that the OCB of employees in PSOs may vary depending on whether or not they are aware of organizational support. Therefore, the following hypothesis is presented.

**Hypothesis 2 (H2).** 
*POS in PSOs has a positive effect on the OCB of employees.*


As mentioned above, POS has an essential influence on customer orientation. CO means providing optimal services by identifying their needs from the perspective of customers [48]. Specifically, CO indicates the attitude, rapid response, and kindness of employees who try to solve customer problems and satisfy their needs [49]. In this study, CO means that employees try to solve problems by identifying their needs and responding quickly through communication efforts with external stakeholders. CO contributes to satisfying customers and building long-term relationships between employees and customers. Berkley and Gupta [50] stated that employees in contact with customers have different attitudes toward treating customers depending on how managers treat them. In the case of service companies, employees with high awareness of POS are likely to try to provide services that meet customer needs to achieve organizational goals.

PSOs aim to realize a healthy life and welfare through sports by providing various sports public services to the people. Employees in PSOs are responsible for forming a customer-oriented mind and contributing to improving the quality of service in public organizations. In this regard, the POS of employees in ski resorts, sports centers, and casinos positively affects CO, showing a significant relationship between the two variables [20,50,51]. The results of these prior studies show that employees of PSOs who highly perceive organizational support are likely to make efforts to provide the best value to customers. Based on these preceding studies, the following hypothesis is presented.

**Hypothesis 3 (H3).** 
*POS in PSOs has a positive effect on the CO of employees.*


### 2.3. Work Engagement (WE), Organizational Citizenship Behavior (OCB), and Customer Orientation (CO)

WE means that employees devote a high level of resources and capabilities emotionally, physically, and cognitively in performing their duties [52]. WE is defined as a positive and fulfilling mental state related to the job, encompassing vitality, dedication, and commitment [53]. In this study, WE is regarded as a valuable thing that contributes to organizational development while enthusiastically enjoying the work of employees. WE encourages employees’ positive energy and work attitude [54] and induces purpose and meaning for the job [55]. Employees with a high level of WE have a solid attitude to take responsibility for their work and show a challenging and active attitude toward new tasks [56].

Meanwhile, WE tends to be immersed in achieving work-related goals given to employees [57]. Employees with WE not only have a high level of emotional attachment to the organization but can also show higher job performance [39]. It can be seen that WE is linked to job performance, which is a crucial factor in creating organizational performance. Among them, WE is closely related to OCB, a representative variable explaining active extra-role behavior within the organization. Kahn [52] noted that employees with WE are willing to use their passion and energy for role performance and extra-role actions. Furthermore, they can consider the organization’s work as part of their area, contributing not only to their own goals but also to achieving the organizational goals [58]. In sports, it was reported that the WE of employees in private and public sports centers had a positive effect on OCB [59,60]. This study proposes the following hypothesis based on existing theoretical and previous empirical studies.

**Hypothesis 4 (H4).** 
*WE in PSOs has a positive effect on the OCB of employees.*


Employees’ work attitudes and roles are considered to be paramount in an organization that must provide customers with high-quality services. WE is a factor dealt with in-depth in service organizations where the role of employees is significant. Employees with a positive attitude, constant effort, and passion for their work will provide better customer service. The higher the WE, the more active the employees can perform their duties and devote themselves to building favorable relationships with customers [61]. In other words, employees with WE perform customer-oriented tasks to satisfy the diverse needs of customers. Recent studies in various fields have also confirmed that WE has a positive effect on CO [55,62,63]. In particular, Son, Han, and Kim [64] mentioned that the WE of a golf caddy has a significant impact on CO, confirming the importance of WE in the sports service industry. On that account, the following hypothesis can be established based on the relationships between variables and the results of previous studies.

**Hypothesis 5 (H5).** 
*WE in PSOs has a positive effect on the CO of employees.*


### 2.4. Mediating Role of Work Engagement (WE)

WE is a significant factor in the organization in terms of sustainable growth and securing its employees’ competitive advantage [65]. Employees who show a high level of WE not only devote themselves to their work but also show a passionate attitude in the informal domain, which greatly affects organizational performance [66,67]. For this reason, WE is used to explain the primary mechanisms by which various personal and organizational factors affect job performance [52]. Recently, miscellaneous studies have been presented in South Korea to verify the mediating effect of WE in the relationship between organizational support and organizational performance. Kim and Cho [68] said that WE had a partial mediating effect on the relationship between coaching leadership and innovation of their supervisors, and Ryu, Kim, and Cha [69] mentioned that WE mediates between job autonomy and job performance. Also, Zhang and Lee [70] showed a partial mediating effect in the impact of Pygmalion leadership, including confidence promotion, human treatment, and task delegation, on job performance. The results of these preceding studies show that WE can be an important factor in explaining the relationship between various antecedent factors, organizational citizenship, and customer orientation. In consequence, the following hypothesis is presented.

**Hypothesis 6 (H6).** 
*WE in PSOs mediates the relationship between POS and OCB of employees.*


**Hypothesis 7 (H7).** 
*WE in PSOs mediates the relationship between POS and CO of employees.*


Based on the hypotheses, the proposed model is shown in Figure 1.

## 3. Methods

### 3.1. Participants and Procedures

This study collected data from 305 employees working for PSOs using the convenience sampling method. PSOs indicate Korea Sports Promotion Foundation (KSPO), Korean Sport & Olympic Committee (KSOC), Athletic organizations, Taekwondo Promotion Foundation, Korea Paralympic Committee (KPC), KSPO & CO. Prior to the distribution of the questionnaire, the purpose of this study was sufficiently explained to each PSO. It was advised in advance to stop the survey immediately or replace it with an online survey, considering that the content and response of the survey could be burdensome in that they reveal the individual’s perception and attitude toward supporting employees of the organization. Specifically, most of them individually responded through paper questionnaires, and 20 people responded via online surveys due to concerns about exposure to personal information and response content. Upon screening the responses for reliability, 57 copies were excluded, such that data from only 248 employees was used for data analysis. According to the list-by-list deletion method suggested by DeSimone and Harms [71], questionnaires in which respondents omitted some responses or in which more than nine questions were consecutive were excluded from data collection. In particular, most of the excluded questionnaires had nine or more of the same responses in a row. The respondents consisted of 177 males (71.4%) and 71 females (28.6%). Also, the average age is in the early–mid-30s (SD = 0.935), and the average number of years of employment is approximately 5 to less than 10 years (SD = 1.400). The specific profile of the respondents is presented in Table 1.

### 3.2. Measures

The instrument for measurement was a questionnaire in which all questions, except for those about demographics, were measured using a 5-point Likert scale (1 = Strongly disagree, 5 = Strongly agree). Before its distribution, a draft consisting of 25 questions was examined for content validity by an expert group formed by one professor, three researchers with Ph.D. degrees, and five employees working for PSOs. To reduce the common method bias in which the correlation between the two variables is more exaggerated than it actually is during the data collection process, the source of the scale for measuring each variable was different, and the order of the questions was partially changed during the research design process. The resulting final questionnaire consisted of 25 questions, including 5 for POS, 5 for WE, 5 for OCB, 5 for CO, and 5 for demographic factors.

POS-related questions consisted of those used in the study of Shanock and Eisenberger [72] and Kim and Lee [73], based on the measurement tool of SPOS (Survey of perceived organizational support) developed by Eisenberger et al. [28]. Questions such as “My organization appreciates my contribution to the organization”, “My organization respects my goals and values”, and “My organization pays attention to my growth and development” were included. The measurement tool for WE was used by Bae et al. [74] based on the research of Schaufeli, Bakker, and Salanova [75]. Questions such as, ‘I enjoy going to work’ and ‘I work passionately in my duties’ were included. To measure OCB, the questionnaire used by Jeong [76] was revised and supplemented based on the five components suggested by Organ [45]. The questionnaires, such as ‘I try to help new employees in the department adapt even if it is not my job’ and ‘I put the interests of the organization before my personal interests within the organization’ were included. Lastly, the questionnaires for CO were adopted by revising items used in Daniel and Darby [77] and Yoon [78]. Questions such as ‘I try to grasp the needs of external stakeholders’ and ‘I try to respond quickly and accurately to questions from external stakeholders’ were included.

## 4. Results

### 4.1. Preliminary Data Analysis

SPSS 26.0 and AMOS 26.0 were used to conduct frequency, confirmatory factor, reliability, correlation, and structural equation model analyses on the collected data. The confirmatory factor analysis was conducted to verify the construct validity for all research variables. Results showed that the model had an appropriate goodness-of-fit (*x*^2^ = 337.474, *df* = 113, *p* < 0.001; CFI = 0.920, TLI = 0.904, and RMSEA = 0.090) because it met the condition of CFI and TLI being 0.90 or higher, and RMSEA being lower than 0.10 suggested by Bagozzi and Dholakia [79] and Woo [80].

Next, standardized factor loadings, construct reliability (CR), and average variance extracted (AVE) values were calculated for convergent validity. As a result, a total of 22 questions were used in this study, excluding 2 questions from WE and 1 question from OCB because they did not meet the value of standardized factor loading (0.50 or higher) suggested by Woo [80]. CR and AVE are distributed between 0.840~0.937 and 0.570~0.751, respectively, which meet the standard (CR = 0.70 or higher and AVE = 0.50 or higher) suggested by Bagozzi and Yi [81] and Fornell and Larcker [82]. Lastly, reliability analysis was conducted using Cronbach’s α to ensure the internal consistency of the measurement instruments. As a result of the analysis, all factors were found to be 0.80 or higher, which means that there was no problem with the reliability of each factor. The final results of confirmatory factor analysis are shown in Table 2.

### 4.2. The Results of Correlation Analysis

Before verifying the hypotheses formulated in this study, a correlation analysis between POS, WE, OCB, and CO was conducted (Table 3). The correlation coefficient between each variable had a significant relationship, and no case exceeded 0.80, confirming no multicollinearity problem [83]. Discriminant validity was well established as the square root of AVE in each latent variable was larger than other correlation values among the latent construct.

### 4.3. The Results of Heterotrait-Monotrait (HTMT) Criterion

Along with correlation analysis, this study conducted an HTMT ratio of correlation to assess discriminant validity (Table 4). Once the value of the HTMT is higher than the threshold, 0.85, it can be concluded that there is a lack of discriminant validity [84]. As a result, since the values are between 0.464 and 0.777, it was confirmed that there was no problem with discriminant validity by HTMT ratio of correlation.

### 4.4. Hypotheses Testing

The hypotheses for the structural model between POS, WE, OCB, and CO were verified through structural equation modeling for which the model’s goodness-of-fit index was acceptable (*x*^2^ = 344.112, *df* = 114, *p* < 0.001; CFI = 0.918, TLI = 0.902, and RMSEA = 0.090). As shown in Table 5 and Figure 2, analysis of the research model showed that the path coefficient of H1 was 0.739 (*t* = 9.868, *p* < 0.001), indicating that POS has a significant positive effect on WE. Next, the path coefficients of H2 and H3 were 0.123 (*t* = 1.305, *p =* 0.192) and −0.152 (*t* = −1.507, *p* = 0.132), indicating that POS significantly has no effects on OCB and CO, respectively. The path coefficients of H4 and H5 were 0.709 (*t* = 6.363, *p* < 0.001) and 0.803 (*t* = 6.299, *p* < 0.001), indicating that WE has significant positive effects on OCB and CO, respectively.

### 4.5. Mediation Effect Testing

As mentioned above, POS of PSOs had a positive effect on WE. Also, WE significantly affected OCB and CO. Since these results meet the prerequisites for verifying the mediating effect presented by Holmbeck [85], the bootstrapping method suggested by Shrout and Bolger [86] was conducted to examine the mediating effect of WE. This method presents a confidence interval (CI) for the measurement of indirect effects, and if the interval does not contain zero, the indirect effect could be considered significant [83]. As shown in Table 6, H6 and H7 were statistically significant because the CI of the path does not contain zero as β = 0.524, 95% Bias-corrected CI [0.348, 0.790], and β = 0.594, 95% Bias-corrected CI [0.411, 0.815], respectively. Thus, WE has been shown to fully mediate the relationship between POS and OCB (H6) and POS and CO (H7).

## 5. Discussion

This study investigates the relationship between POS, WE, OCB, and CO and identifies a mediating role of WE. Based on the results of hypothesis testing, the following discussion is described.

### 5.1. The Relationship between POS and WE

It was found that the POS of employees in PSOs had a positive effect on their WE. Prior studies support the result of this study by stating that the POS of employees in public institutions and social workers has a positive effect on organizational commitment and WE [87,88]. Also, Imran et al. [36] support this study by arguing that when employees feel that they receive support from the organization, they are rewarded with a high level of commitment and participation in their work. This means that PSOs provide special emotional support to their employees. They are more likely to perform their given tasks passionately and happily. As such, the result of this study supports the social exchange theory proposed by Eisenberger et al. [8] that employees who feel recognized by the organization give the organization cooperate and devote themselves to the organization. Imran et al. [36] argue that POS is a fundamental factor in WE, contributing significantly to achieving organizational productivity and sustainability in an uncertain business environment. In this respect, Hekman et al. [89] reported that employees can receive more organizational support when they feel a strong sense of unity. The sense of unity deeply correlates with the value fit between the organization and its employees or their sense of belonging [90,91]. In other words, if PSOs truly acknowledge their employee’s job importance and achievements, they will be attached, committed, and passionate about the organization. Also, PSOs should effectively share visions and values with employees based on mutual trust between labor and management to increase the level of unity.

Additionally, Shore and Shore [92] regarded trust and support from managers, bosses, and colleagues, education and training, promotion, and wage increases as major prerequisites for POS. Similarly, Eisenberger et al. [8] defined interest in job satisfaction, recognition of value for achievements, support for competency development, and support in times of difficulty as a component of POS. In this context, managers’ leadership in PSOs will contribute to employees taking active actions in their jobs by recognizing the organization’s support efforts. For example, authentic leadership of employers influences WE among outcome variables that positively affect the organization [93]. Next, PSOs should pursue the satisfaction of employees’ self-growth needs and organizational growth by strengthening their education and training. In particular, when it is highly job-related and appropriate goal setting is established, it would significantly affect organizational effectiveness [94]. Finally, PSOs should pay attention to appropriate and fair compensation and treatment for their employees’ achievements. A fair evaluation compensation system for employees will increase their trust in the organization and encourage them to perform their duties enthusiastically [74].

### 5.2. The Relationship between POS, OCB, and CO

It was found that POS did not significantly affect both OCB and CO. First, in the relationship between POS and OCB, Lee [95] supports the result of this study by reporting that police officials’ POS had not significantly affected OCB. Yet, this result is contrary to the research of Kim [96] and Rhoades and Eisenberger [10], who found that public officials’ POS has a significant effect on OCB and organizational performance. Several previous studies noted that POS had a positive effect on organizational innovation behavior, creative behavior, and self-directed behavior, which was contrary to the results of this study [97,98,99]. Similarly, it was found that the perception of career support and POS of employees had a significant effect on the formation of customer-oriented attitudes, which contradicted the results of this study [12,100]. This is a different result from the argument of Berkley and Gupta [14] that the attitude toward customers varies depending on the treatment employees receive from managers. In summary, this study shows that the POS of PSOs might not be directly related to their OCB and CO.

The consistent results imply that the improvement of POS within PSOs may have limitations in drawing active, voluntary, and customer-oriented behaviors within the organization at once. According to social exchange theory, an organization and its employees are premised on forming exchange relationships under mutual trust [101]. Employees who benefit from the organization are committed to the organization based on the belief that they are recognized [35]. To illustrate, if an organization supports its employees with salaries, promotions, and emotional recognition, they will respond with effort and loyalty to the organization. Still, in this study, since POS consisted only of emotional support, employees may have judged that POS was somewhat lower than their efforts and contributions. To put it another way, it means that current employees of PSOs do not have enough POS to voluntarily pursue altruistic and customer-oriented behavior within the organization. Therefore, it would be necessary for PSOs to expect balanced reciprocity from employees by paying attention to instrumental and emotional support for them. Furthermore, PSOs should strive to create an environment where they can immerse themselves in their jobs through steady efforts to meet each employee’s needs. Through this, PSOs should consistently increase the level of voluntary motivation of their employees to induce extra-role and customer-oriented behaviors within the organization.

### 5.3. The Relationship between WE, OCB, and CO

The WE of employees in PSOs had a significant effect on OCB. This result was similar to previous studies that indicate that when employees focus on their work, they can show progressive job attitudes or OCB [102,103]. Also, it is in line with the results that WE and OCB have a positive correlation [104,105]. As such, the results of this study imply that if employees have a high mindset to focus and devote themselves to their jobs, they might be more likely to contribute to the organization more actively through extra-role behavior. Christian et al. [58] argued that employees with a high level of WE actively participate in extra-role behavior and activities by efficiently achieving the organization’s goals and appropriately utilizing the resources within the organization. Similarly, employees with WE tend to expand their areas of activity as much as possible within the organization [106]. These voluntary and active activities of employees are in line with the representative characteristics of OCB based on altruism, sincerity, and civic virtue. Therefore, PSOs should create an environment where members can focus on their jobs and take sincere voluntary actions for the organization regardless of price. In particular, OCB can be the basis for forming a healthy organizational culture, so the organization should try to create an environment for the WE of employees.

Next, the WE of employees had a significant effect on CO. This supports the results of prior studies that show that the WE of golf caddies and hotel employees positively affects CO [63,64]. In the public sector, it is reported that the higher the WE of public officials, the higher the probability of having a customer-oriented attitude, supporting the results of this study [107,108]. In other words, employees’ high level of WE can strengthen attitudes to understand and satisfy customers. In this regard, Kim and Moon [109] noted that the high level of WE of employees results in active attitudes and actions in the process of performing their jobs. This active and progressive attitude will contribute to overcoming unexpected situational constraints and forming a customer-oriented attitude that prioritizes customer interests when interacting with customers. Employees with a high level of organizational commitment tend to be more willing to actively engage in and solve customer difficulties [110]. Since employees of PSOs provide sports-related services at the closest point to various stakeholders, active communication and problem-solving capabilities are required. Additionally, spreading performance and positive effects within the organization by increasing employees’ commitment is the ultimate goal that the organization should pursue [111]. Therefore, PSOs should support their members in providing satisfaction to customers by feeling rewarded, proud, and accomplished for their work based on autonomy in performing it. Additionally, PSOs should help their employees psychologically immerse themselves in their work so that they can solve their customers’ problems with expertise.

### 5.4. The Mediating Role of WE

It was found that the WE of employees had a complete mediating effect on the effect of POS on OCB and CO. This result supports the study of Oh [112] that public officials’ WE mediates the relationship between POS and organizational effectiveness. In addition, Jin and Kim [47] support the results of this study by reporting that the WE of employees of social welfare institutions mediates between POS and OCB. As such, the results of this study mean that the effectiveness of organizational support for employees can be maximized when an environment in which they can passionately concentrate on their work is created. These results demonstrated the importance of improving employees’ WE for efficient human resource management.

Kahn [52] argued that employees need a mental state of meaningfulness, safety, and availability to experience a high level of WE. In light of this study, it would be meaningful to use various strategies to strengthen the three factors when supporting employees of PSOs. First, organizations should inspire their employees’ WE through progressive feedback and positive praise [18]. Next, they should create a work environment where employees can experience challenges, autonomy, growth, and satisfaction. Lastly, organizations should make their employees feel available by giving them the belief that they can achieve performance and achievements within the organization. Thus, if PSOs provide supportive feedback to employees and create an active work environment based on trust, they would practice voluntary actions for the sustainable development of the organization and attitudes toward pursuing the satisfaction of external stakeholders.

## 6. Conclusions

This study aims to identify the relationship among POS of employees in PSOs, WE, OCB, and CO and to verify the mediating effect of WE. This study would be meaningful in providing theoretical significance of the relationship between POS of PSOs and variables representing employees’ job attitudes and behaviors. Also, this aims to present strategic and practical implications for efficient human resource management of PSOs. The main results of this study are as follows.

In the first place, POS has a significant positive effect on WE. In the second place, POS significantly has no effects on OCB and CO, respectively. In the third place, WE has significant positive effects on OCB and CO, respectively. Lastly, WE has been shown to fully mediate the relationship between POS, OCB, and POS and CO. The results of this study imply that well-organized support is needed to increase the WE of employees within PSOs. In the same vein, it seems necessary for organizational support to focus on creating conditions in which energy can be put into their role to concentrate on their job. The theoretical and managerial implications of these are as follows.

### 6.1. Theoretical Implications

First, this study is important in that employees of major PSOs in South Korea were selected as subjects for research, and an empirical analysis was conducted to identify their POS. Since most of the previously related studies have been conducted on employees of private sports centers, it might be difficult to apply the results equally to those of the public sector, especially administrative positions. Unlike private organizations that prioritize profit-seeking, this study is meaningful in that it attempted to provide a direction for human resource management strategies of PSOs, which can lead OCB and CO of employees at a time when public demand for public sports services and sports activities as universal welfare increases.

Then, this study revealed that WE plays a complete mediating role in the relationship between POS, OCB, and CO. It was found that psychological mechanisms through WE are very important for organizational effectiveness in support at the organizational level. Therefore, this study will have academic significance in that it has contributed to exploring the positive effects of WE and led to various follow-up studies related to it.

Ultimately, contrary to previous research results, this study showed that POS of PSOs may not directly lead to extra-role and customer-oriented behaviors of employees. It can be assumed that this could result in diverse behaviors that are beneficial to the organization after employees experience positive psychological emotions, including WE. Therefore, this study suggested that there is a need to understand the structural relationship between POS and organizational effectiveness using various mediating variables in the future.

### 6.2. Managerial Implications

First, PSOs need to establish systematic supporting plans to strengthen the POS of employees. Support from supervisors and colleagues is important, but constant efforts to support and motivate employees at the organizational level are needed. For example, specific measures should be prepared for the development of employees’ competencies, such as employee-centered education and training programs, a fair compensation system for performance, and support for resources necessary to achieve goals.

Second, the results of this study show that relatively more attention and effort are needed to strengthen WE to improve organizational effectiveness. In particular, today, organizations need employees with an active attitude toward the organization and work to respond actively to the rapidly changing external environment. Accordingly, PSOs should create a work environment where employees can be autonomously motivated and work passionately. For instance, PSOs need to give employees autonomy in decision-making processes, work schedule adjustment, and establishing a project execution plan.

Finally, PSOs should strive to build a horizontal organizational culture that respects the individuality and creativity of employees so that they can feel emotional stability. Traditionally, PSOs in South Korea have a more bureaucratic organizational structure and a unique hierarchical culture than other public organizations. It might be a good idea to have organizations use English names, initials, or nicknames instead of positions and positions to create a horizontal organizational culture. In addition, it would be necessary to allow employees to freely use the flexible work and work-from-home system if necessary or to create a culture in which they meet 1:1 with senior managers to consider and share the direction of the organization and their growth.

### 6.3. Limitations and Future Research Directions

First, there is a limitation to generalizing the results to every PSO in South Korea, as the subject of this study was only 248 employees. Also, each PSO has a different organizational size, system, and culture. Consequently, future research needs to contribute to establishing differentiated strategies by dividing PSOs into two groups according to the size of the organization and then verifying the differences between groups through multi-group analysis.

In the second place, future research needs to consider various preceding variables that can affect WE. The results of this study suggest that WE is critical in the relationship between POS and organizational effectiveness through a complete mediating role. Thus, follow-up studies need to expand the scope of research by linking with various leading variables, such as job characteristics, job suitability, and leadership.

As a final point, this study investigated the relationship between variables based on the results of data collected at a specific point in time. In particular, a cross-sectional study has a limitation in estimating mediation effects that accurately reflect the temporal antecedent relationships of variables. Due to these problems, a longitudinal study is needed to analyze the results of this study more accurately. Therefore, future research is expected to provide useful information for establishing effective human resource management strategies and improving organizational performance if it analyzes the changes in attitudes and behaviors of employees according to POS of PSOs through longitudinal research.

## Figures and Tables

**Figure 1 behavsci-14-00153-f001:**
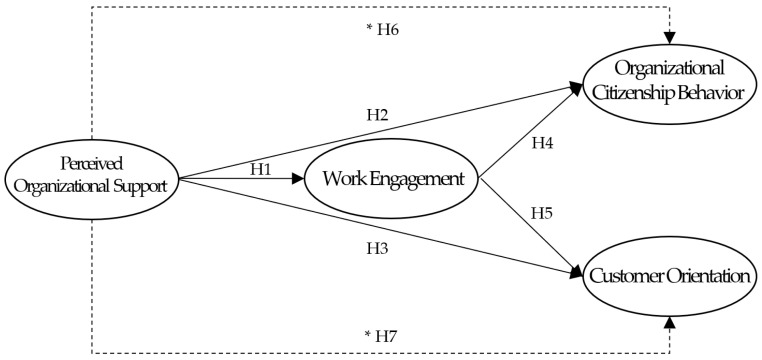
Proposed model. Note: * Verifying mediating effect.

**Figure 2 behavsci-14-00153-f002:**
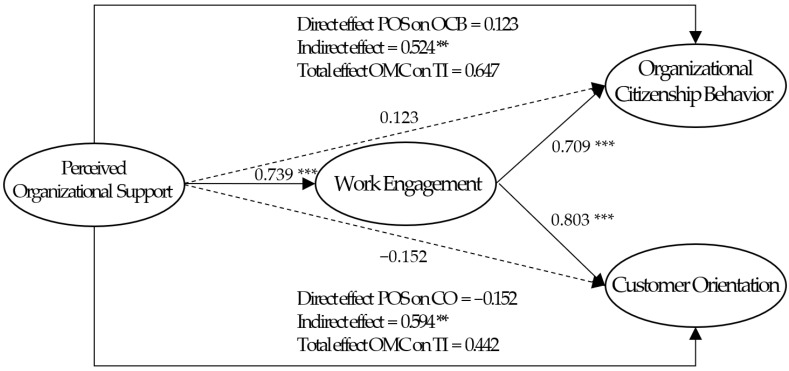
Result for estimates of the model. Note: Figure shows standardized beta values; *** *p* < 0.001, ** *p* < 0.01.

**Table 1 behavsci-14-00153-t001:** Demographic characteristics (*n* = 248).

Category		*n*	%
Gender	Male	177	71.4
	Female	71	28.6
Organization	Korea Sport & Olympic Committee	12	4.8
	Korea Sport Promotion Foundation	76	30.6
	KSPO & CO	38	15.3
	Athletic organization	81	32.7
	Taekwondo Promotion Foundation	24	9.7
	Korea Paralympic Committee	17	6.9
Position	Staff or Senior Staff	73	29.4
	(Assistant) Manager	119	48
	(Deputy) General Manager	20	8.1
	Head of the Department	6	2.4
	Temporary Worker	20	8.1
	Others	10	4

**Table 2 behavsci-14-00153-t002:** The results of confirmatory factor analysis.

Variables and Items	Estimate	SE	CR	AVE	α
**Perceived Organizational Support**					
My organization appreciates my contribution to the organization	0.825		0.934	0.738	0.930
My organization respects my goals and values	0.881	0.065			
My organization pays attention to my growth and development	0.879	0.071			
My organization shows me a lot of interest	0.851	0.065			
My organization is proud of the work I have done	0.833	0.063			
**Work Engagement**					
I enjoy going to work	0.749		0.840	0.637	0.805
I work passionately in my duties	0.718	0.067			
My work is a valuable thing that contributes to organizational development	0.848	0.078			
**Organizational Citizenship Behavior**					
I try to help new employees in the development adapt even if it’s not my job	0.694		0.841	0.570	0.803
I participate in non-work events for the sake of the organization	0.743	0.118			
I put the organization’s interests before my private interests within the organization	0.656	0.116			
I tend to talk about good things about the organization and my colleagues	0.755	0.114			
**Customer Orientation**					
I try to grasp the needs of external stakeholders	0.664		0.937	0.751	0.891
I try to respond quickly and accurately to questions from external stakeholders	0.758	0.097			
I try to solve the problems of external stakeholders	0.854	0.100			
I try to maintain friendly relations with external stakeholders	0.836	0.109			
I frequently try to listen to the opinions of external stakeholders	0.846	0.108			
*x^2^ =* 337.474, *df =* 113, *p* < 0.001, CFI = 0.920, TLI = 0.904, and RMSEA = 0.090					

**Table 3 behavsci-14-00153-t003:** Descriptive statistics and correlation analysis.

Variables	M	SD	1	2	3	4
1. Perceived Organizational Support	3.19	0.87	1			
2. Work Engagement	3.69	0.79	0.649 **	1		
3. Organizational Citizenship Behavior	3.79	0.70	0.553 **	0.630 **	1	
4. Customer Orientation	3.82	0.63	0.424 **	0.571 **	0.554 **	1

*** p* < 0.01.

**Table 4 behavsci-14-00153-t004:** The result of HTMT.

Variables	1	2	3	4
1. Perceived Organizational Support	1			
2. Work Engagement	0.737	1		
3. Organizational Citizenship Behavior	0.638	0.777	1	
4. Customer Orientation	0.464	0.666	0.654	1

**Table 5 behavsci-14-00153-t005:** The results of hypotheses testing.

Path	Estimate	SE	*t*	95% CI	*p*	Result
H1. POS ---> WE	0.739	0.075	9.868	0.499~0.671	0.000	Accepted
H2. POS ---> OCB	0.123	0.081	1.305	0.361~0.529	0.192	Rejected
H3. POS ---> CO	−0.152	0.067	−1.507	0.224~0.388	0.132	Rejected
H4. WE ---> OCB	0.709	0.095	6.363	0.476~0.650	0.000	Accepted
H5. WE ---> CO	0.803	0.085	6.299	0.374~0.539	0.000	Accepted

Note: POS: Perceived organizational support; WE: Work engagement; OCB: Organizational citizenship behavior; CO: Customer orientation.

**Table 6 behavsci-14-00153-t006:** The result of mediation testing.

Path	Direct Effect	Indirect Effect (*p*)	Total Effect	95% CI (Bias-Corrected Bootstrap)	
				Lower	Upper
H6. POS ---> WE ---> OCB	0.123	0.524 (0.003)	0.647	0.348	0.790
H7. POS ---> WE ---> CO	−0.152	0.594 (0.004)	0.442	0.411	0.815

Note: POS: Perceived organizational support; WE: Work engagement; OCB: Organizational citizenship behavior; CO: Customer orientation.

## Data Availability

Data will be provided on demand.

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
