# Peer review of "The Relationship between Perceived Organizational Support, Work Engagement, Organizational Citizenship Behavior, and Customer Orientation in the Public Sports Organizations Context"

_behavsci, 2024, doi:10.3390/bs14030153_

Round 1

Reviewer 1 Report

Comments and Suggestions for Authors

The article studies a very relevant topic in human resource management research, which is the central role of  work engagement in promoting positive outputs such as organizational citizenship behaviors and customer orientation.

The article is well structured and presents the data in an organized and coherent way.

However, I would like to ask why the authors present the data on the age of the sample as well as years of employment divided into categories. The criteria for this division are not presented, nor is this data used throughout the work. There is no reason to present the data in categories, unless they are natural categories (men/women), if there is no defined criterion and the data is not then considered in the analysis and discussion of the data.

Thus, for age and years of employment, only the mean and SD should be presented.

Some of the results, surprisingly, are not in line with the research, such as the non-relationship between POS and OCB and POS and Customer Orientation.

However, in the discussion and conclusions, the authors try to support this data, although more research is certainly needed to understand and explain it.

For these reasons, I think the article can be published with minor alterations.

Reviewer 2 Report

Comments and Suggestions for Authors

Dear authors,

I extend my congratulations to you for your research on the role that POS plays in employees' work-related behaviors and attitudes. I believe that your work constitutes an important contribution to both researchers and practitioners in the fields of organizational behavior and human resource management. Overall, I commend you for the excellent job you have done.

However, I would like to offer two recommendations aimed at enhancing the clarity of your manuscript. Firstly, I suggest enhancing the initial review of the constructs under analysis to provide the reader with a more operational understanding of each variable under scrutiny. Additionally, I recommend presenting the discussion in a more integrated manner, rather than in separate sections, to offer the reader a clearer insight into the psychological processes underlying the relationship between POS and OCB and CO. Furthermore, considering the focus of your study, I believe that the section on the indirect effect of WE should receive greater emphasis.

Once again, congratulations on your work!

Reviewer 3 Report

Comments and Suggestions for Authors

The paper is generally well structured and presented but there are some improvements that are needed before the paper can be acceptable.

1. The abstract needs to be written more informatively.

2. Check all the in text citation as you should not be using & in text citation it should be and.

3. There is no theoretical underpinning which is strange as in quatitative research you begin modeling with the theory and then your contribution.

4. Literature review is mostly dated dan needs to eb updated.

5. The authors need to clearly explain how data was collected. Was there a filter question?

6. Why were 57 copies of the questionnaire discarded as this is a large number and may show some pattern.

7. Are there no control variables as in this kind of study others have controlled for age, gender and tenure?

8. Data is single source but single source bias was not addressed.

9. Data is cross sectional but mediation was tested, see Aguinis et al. (2017) for the criticism.

10. For discriminant validity, the auther should report HTMT ratio.

11. Tables 4 shoudl also include confidence intervals.

12. Discussion needs not enhanced to compare and contrast with the literature after they are updated.

13. Implications also needs more thought to eb specific and actionable.

14. Re-look at teh limitations seriously as they should jeopardize the validity of the current study.

15. Suggestions for future research needs to pint out specific variables or strategy that others can explore.

Comments on the Quality of English Language

Acceptable in most parts of the paper.
